# Clinical and Neurophysiological Follow-Up of Chronic Inflammatory Demyelinating Polyneuropathy Patients Treated with Subcutaneous Immunoglobulins: A Real-Life Single Center Study

**DOI:** 10.3390/brainsci13010010

**Published:** 2022-12-21

**Authors:** Paolo Alonge, Vincenzo Di Stefano, Antonino Lupica, Massimo Gangitano, Angelo Torrente, Antonia Pignolo, Bruna Maggio, Salvatore Iacono, Francesca Gentile, Filippo Brighina

**Affiliations:** Neurology Unit, Department of Biomedicine, Neuroscience, and Advanced Diagnostics (BiND), University of Palermo, 90129 Palermo, Italy

**Keywords:** CIDP, SCIg, cMAP, SNAP, ISS, INCAT, MRC, subcutaneous immunoglobulin

## Abstract

Background: chronic idiopathic demyelinating polyneuropathy (CIDP) is an acquired, immune-mediated neuropathy characterized by weakness, sensory symptoms and significant reduction or loss of deep tendon reflexes evolving over 2 months at least, associated with electrophysiological evidence of peripheral nerve demyelination. Recently, subcutaneous immunoglobulins (SCIg) have been introduced in clinical practice as a maintenance therapy for CIDP; nevertheless, electrophysiological and efficacy data are limited. Methods: to evaluate SCIg treatment efficacy, we retrospectively reviewed data from 15 CIDP patients referring to our clinic, receiving SCIg treatment and who performed electrophysiological studies (NCS) and clinical scores (MRC sumscore, INCAT disability score and ISS) before starting the treatment and at least one year after. Results: NCS showed no significant changes before and during treatment for all the nerves explored. Clinical scores did not significantly change between evaluations. Correlation analysis evidenced a positive correlation of cMAPs distal amplitude with MRC sumscore and a trend of negative correlation with the INCAT disability score. Conclusions: SCIg maintenance therapy preserves nerve function in CIDP with a good efficacy and safety. Treatment effectiveness can be assessed with ENG, which represents a useful instrument in the follow-up and prognostic assessment of CIDP.

## 1. Introduction

Chronic idiopathic demyelinating polyneuropathy (CIDP) is an acquired, immune-mediated polyradiculoneuropathy evolving over 2 months at least [1]. The typical form is characterized by sensory symptoms (e.g., paresthesia, sensory loss), distal muscle weakness and reduced or absent deep tendon reflexes, with a distal and symmetrical involvement that progresses proximally [2]. Cranial nerves and the autonomic system are usually spared in CIDP. However, there are also atypical forms in which some of the classical symptoms are absent (e.g., motor CIDP or sensory CIDP) or an asymmetrical or focal involvement is observed [3]. Electrophysiological findings play a key role in the diagnosis and monitoring of CIDP: according to the EAN/PNS (European Academy of Neurology/Peripheral Nerve Society) diagnostic criteria, the demonstration of peripheral nerve demyelination in two or more nerves is required for a defined diagnosis [4]. Electrophysiological variables have been also extensively used in clinical trials to evaluate the response to treatment and the progression of the disease [5,6].

There are several therapeutic options for CIDP, which include intravenous immunoglobulin (IVIG), plasma exchange (PEX) and glucocorticoids. After an induction therapy, most patients require a maintenance therapy with periodic IVIG administration, PEX procedures or immunodepressants (steroids, rituximab) to prevent relapses and progression [4,7,8]. Subcutaneous immunoglobulin therapy (SCIg) has been used as an alternative to IVIG in primary immunodeficiencies for over thirty years. Compared to IVIG, which are administered every 3–4 weeks, SCIg are administered in smaller doses; hence, the frequency of administration is higher (once or twice weekly). Evidence shows that, while the SCIg efficacy is similar to IVIG, patients usually report a lower incidence of side effects (e.g., headache, local reactions in injection site, renal and cardiac impairment) and a better quality of life; this is attributed to the lower peak serum dose reached by SCIg compared to IVIG (61%); another advantage is that SCIg therapy does not require an intravenous access. Hence, SCIg is commonly administered at home [9,10]. Recently, SCIg has been introduced in clinical practice as a maintenance therapy even for CIDP; nevertheless, data on the efficacy of SCIg and electrophysiological data are limited.

## 2. Materials and Methods

### 2.1. Study Procedures

The study was conducted in accordance with the Declaration of Helsinki and approved by the Ethics Committee Palermo I, University of Palermo (Protocol code 07/2020; 13 July 2020). In this study, we present a retrospective evaluation of the efficacy of SCIg treatment in a population of CIDP patients using electroneurography (ENG) and clinical scores.

### 2.2. Patient Demographics


*We reviewed data from patients referring to our clinic (“Policlinico Paolo Giaccone di Palermo” –“Centro per la Diagnosi e Cura della Malattie Neuromuscolari Rare”) from January 2014 to September 2022.*



*Inclusion criteria:*



*Patients accessing our clinic were enrolled in the presence of the following criteria:*


−
*Age >18 years;*
−
*Diagnosis of definite CIDP according to the EAN/PNS 2021 criteria;*
−
*Treatment with SCIg;*
−
*Evaluation with apposite clinical scales (INCAT, ISS, MRC) and nerve conduction studies.*
−
*Exclusion criteria:*
−
*Lack of infomed consent to participation;*
−
*Diagnosis of probable or possible CIDP according to the EAN/PNS 2021 criteria;*
−
*Lack of response to IVIg.*


### 2.3. Clinical Assessment

The inflammatory neuropathy cause and treatment (INCAT) disability score is calculated by summing a score measuring arm impairment (0 = no upper limb problems; 5 = inability to use either arms for purposeful movements) and another measuring leg impairment (0 = walking unaffected; 5 = restricted to wheelchair). The INCAT score can range from 0 (no disability) to 10 (maximum disability) [11].

Similarly, the INCAT sensory sumscore (ISS) evaluates sensory impairment by measuring pinprick sensation, vibration sensation and two-point discrimination at arms and legs. It is calculated by adding scores obtained by all four limbs and it ranges from 0 (normal sensation) to 20 (severe sensory deficit) [12].

The Medical Research Council (MRC) sumscore measures strength by applying the MRC 5-point system (0 = no movement; 5 = movement completed against full resistance) to six muscle groups (abduction of the arm, flexion of the forearm, extension of the wrist, flexion of the leg, extension of the knee, dorsal flexion of the foot) of both sides. It ranges from 0 (minimum strength) to 60 (maximum strength) [13].

The INCAT disability score, the ISS and the MRC sumscore were developed specifically to evaluate patients affected by inflammatory polyneuropathies and have been used in several clinical trials to estimate the efficacy of treatments and the clinical progression of the disease over time.

### 2.4. Nerve Conduction Studies (NCS)

Sensory nerve action potentials (SNAPs) and compound muscle action potentials (CMAPs) were recorded, analyzing distal latencies (dL), negative peak amplitudes (dA) and conduction velocities (CV). We investigated median and ulnar nerves for the upper limbs and peroneal, tibial and sural nerves for the lower limbs, according to standard procedures (i.e., bipolar surface stimulating electrodes delivering rectangular pulses 0.1–0.5 ms in duration with recording electrodes placed over the recording site, with a ground electrode placed between the recording electrodes and stimulation site). In particular, the study protocol was defined as follows:

For upper-limb SNAPs: stimulation at wrist and registration from II digit (medial nerve) and V digit (ulnar nerve);

For upper-limb CMAPs: stimulation at the wrist and elbow and recording from abductor pollicis brevis (APB) for median nerve; stimulation at the wrist and elbow 4 cm distal from the medial epicondyle of the humerus and recording from the abductor digiti minimi (ADM) muscle for the ulnar nerve;

For lower-limb SNAPs: stimulation at the posterior-lateral calf, recording from the lateral malleolus for the sural nerve;

For lower-limb CMAPs: stimulation at the medial malleolus and popliteal fossa, recording from the abductor hallucis brevis (AHB) muscle for the tibial nerve; stimulation at the anterior ankle and popliteal fossa, recording from the extensor digitorum brevis (EDB) muscle for the peroneal nerve.

### 2.5. Statistical Analysis

Neurophysiological variables (continuous) and clinical scores (discrete) were compared using the Mann–Whitney test to detect changes between the evaluations before and during therapy. Correlations between clinical scores and neurophysiological variables (cMAPs and SAPs distal amplitude) were evaluated using Pearson’s r value. Significance was set at 0.05 for all the analyzed variables. Data are presented as a median ± interquartile range, except for correlations, which are presented as a mean value ± standard deviation. All the analyses were performed with JASP (version 0.16.2; computer software).

## 3. Results

Out of 19 patients, 3 were lacking a baseline electrophysiological evaluation and 1 did not reach a 1-year follow-up. Fifteen patients were included in the final analysis (9 males, 56 ± 13 years; see Table 1 for population characteristics). Thirteen patients (87%) showed a typical CIDP phenotype, while two showed an atypical pattern (motor CIDP). Five patients out of fifteen did not have an evaluation with the mentioned clinical scales before and/or after treatment start; therefore, analysis on clinical scores and correlations were performed on the remaining 10 patients. Evaluations during SCIg treatment (both clinical and neurophysiological) were performed after a median interval of 16 months (IQ 13–19) from the start. Median time between ENG evaluations was 37 months (IQ range 29–42). The median time between clinical evaluations was 35 months (IQ range 10–20; see Figure 1 for histogram of follow-up time). Before starting SCIg as maintenance therapy, nine patients (60%) were treated with prednisone; nine patients (60%) received IVIg as a maintenance therapy; one patient (6%) was treated with cyclophosphamide; and one (6%) with azathioprine.

At baseline, the median nerve (both motor and sensitive) was tested in 60% of patients, the ulnar (both motor and sensitive) nerve in 60%, the peroneal motor nerve in 60%, the tibial nerve in 73.3% and the sural nerve in 20%. Repeat testing was conducted in 100% of patients for all the nerves, except for the sural nerve, which was tested in 66% of the patients at follow-up.

NCS showed no significant changes before and during treatment for all the nerves explored (Table 2); a trend of worsening was observed for dA (11.9 vs. 4.3 uV; *p* = 0.078) and CV (44.46 vs. 32.13 m/sec; *p* = 0.17) registered from the right sensitive median nerve, while CV registered from the right peroneal nerve improved at follow-up (29.82 vs. 44.78 m/sec; *p*= 0.15).

The median scores before the start of the therapy were 3 (IQ range 3–4) for the INCAT disability score, 10 (IQ range 8–10) for ISS and 54 (IQ range 47–58) for the MRC sumscore. The scores did not significantly change at follow-up (Table 3).

Correlation analysis (Table 4) evidenced a positive correlation of cMAPs distal amplitude with an MRC sumscore (r = 0.2; *p* = 0.05) and a trend of negative correlation with the INCAT disability score (r = −0.156; *p* = 0.15).

## 4. Discussion

Only a few studies reported electrophysiological data of CIDP patients undergoing SCIg treatment. The PATH study, which is the largest trial to evaluate SCIg efficacy in CIDP, reported no significant changes in nerve conduction variables after six months in 115 patients, equally divided in two treatment regimens (0.2 g/kg/week vs. 0.4 g/kg/week), while the placebo group (57 patients) showed a slight worsening of proximal motor latencies and conduction velocities in median, ulnar and peroneal motor nerves. Cirillo et al. reported how SCIg therapy is effective in preserving nerve function in the long term in a population of 14 patients, which also showed an improvement of CMAP amplitude and CVs after 48 months of treatment [14,15,16].

Our data provide additional support that SCIg maintenance therapy is effective in preventing nerve function deterioration in CIDP patients, confirming the findings of the aforementioned studies. The absence of cMAP amplitude improvement in our population could be due to the shorter median follow-up time after the start of therapy compared to the study of Cirillo et al. However, an improvement was seen in single patients (see Figure 2; a block conduction on motor median nerve resolved after treatment in patient 001). Furthermore, we confirmed the presence of a positive correlation between cMAPs amplitude and MRC sumscore, suggesting that ENG variables could hold a role as prognostic factors to estimate treatment efficacy and duration time.

There were no differences between patients with typical and atypical CIDP phenotype in our population. However, the number of patients with atypical characteristics in our population was too small to draw conclusions from our findings. Considering that atypical phenotypes are reported to respond poorly to immunoglobulin treatment [17], further studies are required to investigate whether the efficacy of SCIg treatment changes in atypical CIDP.

A relevant strength of this study is the long follow-up of our cohort of CIDP patients compared to previous studies. Indeed, prolonged therapy with SCIg was safe and provided stable disease burden and neurophysiological data. Moreover, an improvement in the MCV of peroneal nerves support the idea that the demyelinating process and inflammation in CIDP recede from this treatment; also, stable CMAP amplitude on motor nerves confirm that no more significant axonal loss happens during SCIg treatment in CIDP. Another relevant point is the use of neurophysiological variables to assess treatment efficacy. Despite being cheap, easily reproducible and easy to perform, ENG has been seldom used as a long-term follow-up technique in clinical studies. We suggest that neurophysiological examination could provide more detailed information on SCIg treatment efficacy compared to clinical scores.

Our study has some limitations; first, the small size of the population analyzed; second, the lack of a homogenous protocol of ENG testing and clinical follow-up among patients, which reduces the significance of our data; and third, the absence of a control group (i.e., with IVIg administration).

## 5. Conclusions

Our data strengthen the evidence on the efficacy of SCIg maintenance therapy in CIDP. Indeed, no patients presented a worsening of symptoms during maintenance treatment with SCIg and there was good safety. Nerve conduction studies are a useful instrument not only in the diagnostic process, but also in the follow-up and prognostic assessment of CIDP.

## Figures and Tables

**Figure 1 brainsci-13-00010-f001:**
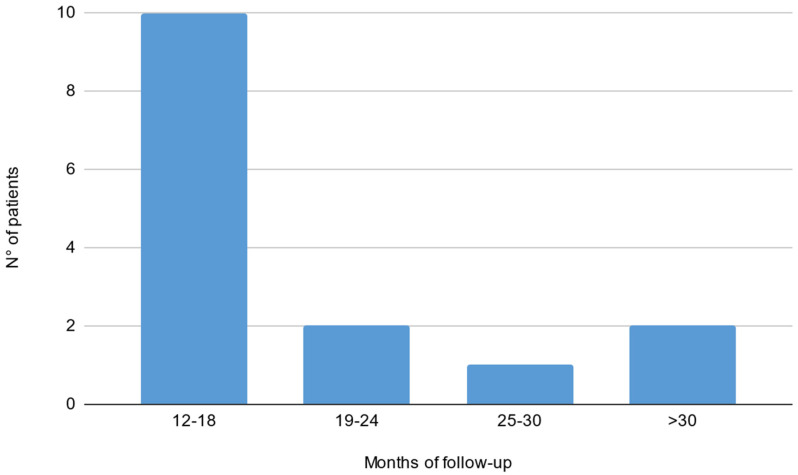
Histogram of follow-up time reported in months in our cohort of CIDP patients treated with SCIg.

**Figure 2 brainsci-13-00010-f002:**
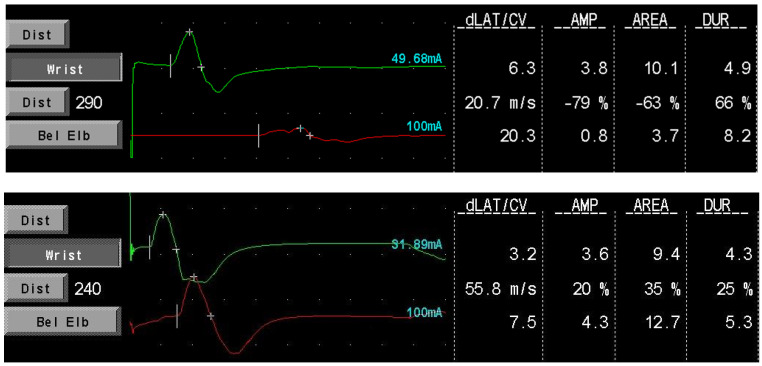
Patient 001 motor medial nerve conduction before (**upper image**) and during (**lower image**) SCIg therapy. The conduction block resolves with treatment. dLAT: distal latency; CV: conduction velocity; AMP: amplitude (mV); DUR: duration (msec).

**Table 1 brainsci-13-00010-t001:** Clinical data in our cohort of CIDP patients treated with SCIg. Chronic inflammatory demyelinating polyneuropathy, CIDP; intravenous immunoglobuline, IVIg; subcutaneous immunoglobulins, SCIg.

Patient Code	Age (Years)	Sex	Duration of Disease (Months)	CIDP Phenotype	Previous Maintenance Therapy	Duration of SCIg Treatment (Months)	SCIg Dosage (Monthly)	Other Conditions
001	64	M	90	Motor CIDP	IVIg	84	60 g	Diabetes mellitus Osteopenia
002	59	F	21	Typical	Prednisone	19	80 g	-
003	52	F	204	Motor CIDP	Prednisone IVIg	33	80 g	-
004	74	M	31	Typical	Prednisone IVIg	13	120 g	Atrial fibrillation
005	80	M	336	Typical	IVIg cyclophosphamide	16	30 g	-
006	48	F	20	Typical	Prednisone	15	60 g	-
007	48	M	21	Typical	Prednisone	18	60 g	-
008	44	M	72	Typical	Prednisone	13	80 g	-
009	44	F	36	Typical	Azathioprine	14	80 g	-
010	61	M	48	Typical	IVIg	13	80 g	-
011	48	M	20	Typical	Prednisone	17	120 g	-
012	73	M	96	Typical	IVIg	30	100 g	Peripheral arterial disease, diabetes mellitus
013	60	M	84	Typical	Prednisone IVIg	13	80 g	Dyslipidemia
014	73	F	76	Typical	IVIg	15	60 g	HCV-related hepatopathy, osteoporosis
015	55	F	50	Typical	Prednisone IVIg	21	60 g	-

**Table 2 brainsci-13-00010-t002:** Electrophysiological data. Time 1: pre-SCIg; time 2: during SCIg; N: number of patients; Dx: right nerve; Sn: left nerve; dL: distal latency; dA: distal amplitude; SAP: sensory action potential; cMAP: compound motor action potential; CV: conduction velocity; SD: standard deviation.

	Time	N	Mean	SD	*p*
SAPs median Dx dL (ms)	1	9	2.319	1.122	0.774
	2	10	2.340	1.845	
SAPs median Dx dA (uV)	1	9	11.867	15.890	0.078
	2	10	4.370	7.170	
SAPs median Dx CV (m/s)	1	9	44.467	20.205	0.175
	2	10	32.130	23.677	
cMAPs median Dx dL (ms)	1	10	4.640	2.118	0.291
	2	12	5.342	1.718	
cMAPs median Dx dA (mV)	1	10	4.430	2.113	0.339
	2	12	3.692	2.190	
cMAPs median Dx CV (m/s)	1	10	39.540	15.752	0.947
	2	12	39.942	12.487	
SAPs median Sn dL (ms)	1	6	2.000	1.698	0.810
	2	9	2.367	2.170	
SAPs median Sn dA (uV)	1	6	2.133	1.900	0.904
	2	9	7.022	11.782	
SAPs median Sn CV (m/s)	1	6	29.583	25.443	0.626
	2	9	24.211	25.225	
cMAPs median Sn dL (ms)	1	9	5.011	2.913	0.837
	2	10	4.650	1.828	
cMAPs median Sn dA (mV)	1	9	4.078	2.100	0.513
	2	10	5.260	3.364	
cMAPs median Sn CV (m/s)	1	9	37.189	15.430	0.842
	2	10	40.250	14.481	
SAPs ulnar Dx dL (ms)	1	6	1.797	1.059	0.301
	2	10	2.367	1.603	
SAPs ulnar Dx dA (uV)	1	6	16.933	20.476	0.703
	2	10	9.370	11.741	
SAPs ulnar Dx CV (m/s)	1	6	44.150	23.869	0.444
	2	10	32.360	25.265	
cMAPs ulnar Dx dL (ms)	1	9	3.452	1.896	0.250
	2	9	4.100	1.429	
cMAPs ulnar Dx dA (mV)	1	9	4.956	2.220	0.690
	2	9	4.511	2.723	
cMAPs ulnar Dx CV (m/s)	1	9	45.267	20.194	0.354
	2	9	40.011	14.217	
SAPs ulnar Sn dL (ms)	1	4	2.100	1.490	0.864
	2	8	2.357	1.221	
SAPs ulnar Sn dA (uV)	1	4	3.250	2.575	0.865
	2	8	8.588	14.304	
SAPs ulnar Sn CV (m/s)	1	4	33.025	23.085	1.000
	2	8	34.112	23.129	
cMAPs ulnar Sn dL (ms)	1	5	5.320	3.214	0.224
	2	9	3.644	1.696	
cMAPs ulnar Sn dA (mV)	1	5	2.800	2.623	0.317
	2	9	4.456	2.706	
cMAPs ulnar Sn CV (m/s)	1	5	32.160	17.373	0.257
	2	9	42.889	14.089	
cMAPs peroneal Dx dL (ms)	1	10	7.530	5.707	0.967
	2	9	6.078	2.621	
cMAPs peroneal Dx dA (mV)	1	10	1.990	1.959	0.870
	2	9	1.756	1.490	
cMAPs peroneal Dx CV (m/s)	1	10	29.820	14.193	0.156
	2	9	44.789	21.443	
cMAPs peroneal Sn dL (ms)	1	5	4.160	3.010	0.881
	2	3	5.233	4.546	
cMAPs peroneal Sn dA (mV)	1	5	1.480	1.462	1.000
	2	3	1.567	1.845	
cMAPs peroneal Sn CV (m/s)	1	5	33.940	21.367	0.453
	2	3	24.533	21.804	
cMAPs tibial Dx dL (ms)	1	11	5.709	3.901	0.526
	2	10	6.160	3.382	
cMAPs tibial Dx dA (mV)	1	11	3.818	3.919	0.526
	2	10	4.630	8.823	
cMAPs tibial Dx CV (m/s)	1	11	33.318	16.021	0.549
	2	10	26.500	20.007	
cMAPs tibial Sn dL (ms)	1	6	5.717	2.927	0.470
	2	6	3.600	3.109	
cMAPs tibial Sn dA (mV)	1	6	3.133	3.840	0.378
	2	6	1.733	2.229	
cMAPs tibial Sn CV (m/s)	1	6	49.817	24.108	0.229
	2	6	27.183	24.222	
SAPs sural Dx dL (ms)	1	3	2.200	2.066	1.000
	2	2	1.450	2.051	
SAPs sural Dx dA (uV)	1	3	9.400	10.054	0.554
	2	2	2.650	3.748	
SAPs sural Dx CV (m/s)	1	3	29.767	30.003	1.000
	2	2	25.850	36.557	
SAPs sural Sn dL (ms)	1	2	1.350	1.909	1.000
	2	3	1.767	1.537	
SAPs sural Sn dA (uV)	1	2	6.000	8.485	1.000
	2	3	7.333	6.351	
SAPs sural Sn CV (m/s)	1	2	22.200	31.396	0.554
	2	3	35.033	30.679	

**Table 3 brainsci-13-00010-t003:** Clinical scores. Time 1: pre-SCIg; time 2: during SCIg; MRC: Medical Research Council sumscore; ISS: INCAT sensory scale; SD: standard deviation.

	Time	N	Mean	SD	*p*
MRC	1	9	53.111	6.660	0.85
	2	9	51.889	7.688	
ISS	1	9	8.444	3.609	0.32
	2	9	6.111	5.183	
INCAT disability score	1	9	2.667	1.803	0.36
	2	9	3.333	2.121	

**Table 4 brainsci-13-00010-t004:** Pearson’s correlations.

Variable		SAPs dA	cMAPs dA
MRC sumscore	Pearson’s r	-	0.21
	*p*-value	-	0.05
ISS	Pearson’s r	−0.05	-
	*p*-value	0.78	-
INCAT disability score	Pearson’s r	−0.12	−0.15
	*p*-value	0.54	0.15

SAP: sensory nerve action potential; cMAP: compound muscle action potential; dA: distal amplitude.

## Data Availability

Data are available from the corresponding author upon a reasonable request.

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
