# Peer review of "Clinical and Neurophysiological Follow-Up of Chronic Inflammatory Demyelinating Polyneuropathy Patients Treated with Subcutaneous Immunoglobulins: A Real-Life Single Center Study"

_brainsci, 2022, doi:10.3390/brainsci13010010_

Round 1

Reviewer 1 Report

In general, the work is acceptable although it is not something that can be considered advanced in terms of new technology.

Title

The title of manuscript is well and competently.

Abstract

It is well defined. However, the objective and aim of the study is not presented clearly. 

Introduction

Add other markers of a good response to immunoglobulin treatment. 

Methodology

Adding information on previous treatment. 

Figures

All figures must be clear

Tables 

Tables should be added in suitable section

DISCUSSION

Discuss that other markers contributed to a positive response to treatment, or just SCIg therapy alone. 

Conclusion

Consider revising whole conclusion for grammar and therefore a better readability.

Author Response

Dear Reviewer 1,

Thanks for your comments. We would like to submit our revised version of the manuscript for possible publication in your Brain Science.

In general, the work is acceptable although it is not something that can be considered advanced in terms of new technology.

R: Authors would like to thank the Reviewer for the time spent evaluating our manuscript. We followed all his/her suggestions, and we believe that the paper is now improved in clarity and intelligibility.

Title

The title of manuscript is well and competently.

Abstract

It is well defined. However, the objective and aim of the study is not presented clearly. 

R: The abstract was modified to clarify the aim of the study. Thank you for the observation. 

Introduction

Add other markers of a good response to immunoglobulin treatment. 

R: We considered INCAT disability scores, ISS and nerve conduction studies. 

Methodology

Adding information on previous treatment. 

R: Information about previous treatments was added on table 1 and in the Results. 

Figures

All figures must be clear

R: We explicated unit of measures for time.

Tables 

Tables should be added in suitable section

R: we thank the reviewer for these suggestions. We modified font and added a caption. According to the MDPI format tables were added in the nearest point of the first citation in the text.

DISCUSSION

Discuss that other markers contributed to a positive response to treatment, or just SCIg therapy alone. 

R: Most patients achieved stable disease with SCIg alone.

Conclusion

Consider revising whole conclusion for grammar and therefore a better readability

R: We revised the conclusion according to your suggestion. 

Hoping for positive feedback we look forward to hearing from you soon.

Kind regards,

Paolo Alonge

Reviewer 2 Report

The paper presented to me for review, "Clinical and neurophysiological follow-up of CIDP patients treated with SCIg: a single center experience," is a study on the treatment of 15 CIDP patients with subcutaneous immunoglobulins. This is still observed in clinical trials as a form of IvIg treatment. In a retrospective study, the authors showed that SCIg maintenance therapy can maintain a good condition in the peripheral nerves, which they confirmed with an ENG study.

Although this is a small-group study and should be called a "pilot study," I believe it adds knowledge to the treatment of CIDP. 

In the discussion, I would suggest referring to the article https://pubmed.ncbi.nlm.nih.gov/35008604/ and discussing whether among the patients discussed in the paper, different clinical phenotypes of CIDP had an impact on the response after treatment. 

Author Response

Dear Reviewer 2,

Thanks for your comments. We would like to submit our revised version of the manuscript for possible publication in your Brain Science.

The paper presented to me for review, "Clinical and neurophysiological follow-up of CIDP patients treated with SCIg: a single center experience," is a study on the treatment of 15 CIDP patients with subcutaneous immunoglobulins. This is still observed in clinical trials as a form of IvIg treatment. In a retrospective study, the authors showed that SCIg maintenance therapy can maintain a good condition in the peripheral nerves, which they confirmed with an ENG study.

Although this is a small-group study and should be called a "pilot study," I believe it adds knowledge to the treatment of CIDP. 

In the discussion, I would suggest referring to the article https://pubmed.ncbi.nlm.nih.gov/35008604/ and discussing whether among the patients discussed in the paper, different clinical phenotypes of CIDP had an impact on the response after treatment. 

R: Authors would like to thank the Reviewer for the time spent evaluating our manuscript and the precious suggestions. We modified the discussion according to the Reviewer's suggestion. We believe that these observations have improved our paper. 

Hoping for positive feedback we look forward to hearing from you soon.

Kind regards,

Paolo Alonge

Reviewer 3 Report

The main question addressed by the research is about. SCIg in CIDP. The approval for SCIg in CIDP was in 2019. The study is relevant because only small studies and clinical trials were reported. The topic is not a novelty, but there are less than 1000 individuals who have already reported with SCIg treatment and CIDP.

Compared with other published material, it is effective in European countries and provide a better understanding of the electrophysiologic outcomes. The manuscript is well-written, clean and easy to read. But, the English need to be politely revised.

The authors describe an approved drug for CIDP. The conclusions are supported by evidence. And the authors address the EMG outcomes of SCIg in individuals with CIDP.

Minor point

1. Avoid abbreviations in the title. E.g., SCIg

Author Response

Dear Reviewer 3,

Thanks for your comments. We would like to submit our revised version of the manuscript for possible publication in your Brain Science.

The main question addressed by the research is about. SCIg in CIDP. The approval for SCIg in CIDP was in 2019. The study is relevant because only small studies and clinical trials were reported. The topic is not a novelty, but there are less than 1000 individuals who have already reported with SCIg treatment and CIDP.

Compared with other published material, it is effective in European countries and provide a better understanding of the electrophysiologic outcomes. The manuscript is well-written, clean and easy to read. But, the English need to be politely revised.

The authors describe an approved drug for CIDP. The conclusions are supported by evidence. And the authors address the EMG outcomes of SCIg in individuals with CIDP.

Minor point

  1. Avoid abbreviations in the title. E.g., SCIg

R: Authors would like to thank the Reviewer for the time spent evaluating our manuscript. We modified the title according to his/her suggestion. The paper is about a real-life experience, as the drug is approved since 2019. We agree with the reviewer that these evidences are important in a real-life setting to support a wider use of SCIg in CIDP. Our study also aims to underline neurophysiological aspects in a population with a wider follow-up.

Hoping for positive feedback we look forward to hearing from you soon.

Kind regards,

Paolo Alonge

Reviewer 4 Report

Dear authors, thank you for the conducted study. Please see the comments

Minor concerns

1.       The title. Please revise. Abbreviations in the title, especially not common (as MI—myocardial infarction or CAD—coronary artery disease, etc.) are not desirable. Probably, you can add ‘the results of the retrospective single-center study' or indicate the follow-up.

2.       Indicate the information about clinical study registration and data on ethics committee. Give the data when and where the study was conducted.

3.       Please describe the inclusion and exclusion criteria.

4.       Give information on whether the patients received other treatment (IVIG administration or glucocorticoids?) before switching to SCIgs.

5.        Is it possible to add the data to form the control group (possibly with IVIG administration). If not, the lack of control should be mentioned in the Limitations of the study.

Author Response

Dear Reviewer 4,

Thanks for your comments. We would like to submit our revised version of the manuscript for possible publication in your Brain Science.

Dear authors, thank you for the conducted study. Please see the comments

Minor concerns

  1.       The title. Please revise. Abbreviations in the title, especially not common (as MI—myocardial infarction or CAD—coronary artery disease, etc.) are not desirable. Probably, you can add ‘the results of the retrospective single-center study' or indicate the follow-up.
  2.       Indicate the information about clinical study registration and data on ethics committee. Give the data when and where the study was conducted.
  3.       Please describe the inclusion and exclusion criteria.
  4.       Give information on whether the patients received other treatment (IVIG administration or glucocorticoids?) before switching to SCIgs.
  5.       Is it possible to add the data to form the control group (possibly with IVIG administration). If not, the lack of control should be mentioned in the Limitations of the study

R: Authors would like to thank the Reviewer for the time spent on our manuscript. We followed all his/her suggestions, and we believe that the paper improved in clarity and intelligibility.

Hoping for positive feedback we look forward to hearing from you soon.

Kind regards,

Paolo Alonge